# DOMAIN KNOWLEDGE INFUSED CONDITIONAL GENERATIVE MODELS FOR ACCELERATING DRUG DISCOVERY

## ABSTRACT

The role of Artificial intelligence (AI) is growing in every stage of drug development. Nevertheless, a major challenge in drug discovery AI remains: Drug pharmacokinetic (PK) and Drug-Target Interaction (DTI) datasets collected in different studies often exhibit limited overlap, creating data overlap sparsity. Thus, data curation becomes difficult, negatively impacting downstream research investigations in high-throughput screening, polypharmacy, and drug combination. We propose xImagand-DKI, a novel SMILES/Protein-to-Pharmacokinetic/DTI diffusion model capable of generating an array of PK and DTI target properties conditioned on SMILES and protein inputs that exhibit data overlap sparsity. We infuse additional molecular and genomic domain knowledge from the Gene Ontology and molecular fingerprints to further improve our model performance. We show that xImagand-DKI generates synthetic PK data that closely resemble real data univariate and bivariate distributions, and can adequately fill in gaps among PK and DTI datasets. As such, xImagand-DKI is a promising solution for data overlap sparsity and may improve performance for downstream drug discovery research tasks. Our code and data are available open-source [1].

## 1 INTRODUCTION

Artificial intelligence (AI) is set to substantially reduce the $2-3 billion dollars and 10-15 years typically required to bring a drug candidate to market (Kim et al., 2021; Wouters et al., 2020). Fewer than 10% of drug candidates successfully reach the market (Wouters et al., 2020), with the vast majority failing in clinical development due to safety and lack of or insufficient activity (Paul et al., 2010). AI is gaining momentum in drug discovery by enabling innovative preclinical approaches, including target selection and identification (Murmu & Győrffy, 2024), drug repurposing (Thafar et al., 2022; Park & Cho, 2025), drug-target interactions (DTI) (Lian et al., 2021), drug property prediction (Kim et al., 2021), de novo generation (Vignac et al., 2023; Hu et al., 2024), and synthetic data generation (Hu et al., 2025).

These advances in AI-driven drug discovery has been fueled by ongoing efforts to promote open access to data for AI training and testing (Huang et al., 2021; Brown et al., 2019; Gaulton et al., 2017). Despite the growing availability of diverse datasets, limited overlap among them presents challenges for research questions that require data integration from multiple datasets (Scoarta et al., 2023). Given that data collection for drug discovery through assay panels is both expensive and time-consuming, synthetic drug discovery data emerges as a promising alternative solution.

Recent advances in AI for drug discovery have leveraged Denoising Diffusion Probabilistic Models (DDPMs) (Jonathan et al., 2020), a new class of diffusion models capable of generating ligand structures (Guo et al., 2023; Vignac et al., 2023; Wu et al., 2022; Igashov et al., 2022). Emerging research has demonstrated that diffusion models can also generate pharmacokinetic (PK) properties (Hu et al., 2025), and when integrated into a ligand diffusion pipeline (Hu et al., 2024). However, sequence-based molecular and biological representations, such as SMILES and amino acid sequences, alone are likely not sufficient in fully capturing the complexity of natural entities like drug molecules, proteins, and omics data. By fusing multiple views of the same molecule or profile, multi-view representation approaches for molecules (Suryanarayanan et al., 2025) and omics profiles (Ma et al., 2024) can yield unified representations with enhanced predictive power.

---

[1]TBD

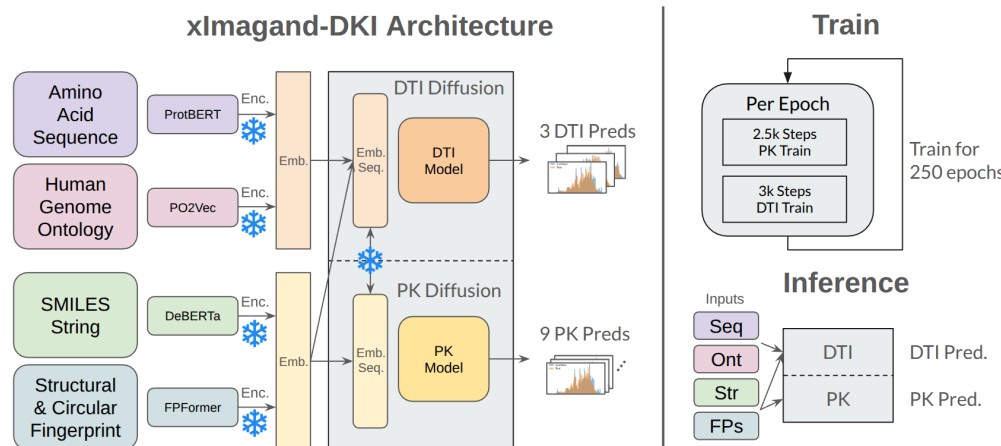

Figure 1: The xImagand-DKI architecture, training, and inference methodology. Embeddings for proteins and SMILES are generated using ProtBERT and DeBERTa, respectively. Protein knowledge infusion from the Gene Ontology knowledge base is generated using PO2Vec, and SMILES knowledge infusion from fingerprints is generated using FPFormer. The model undergoes 2.5k PK training steps and 3k DTI training steps every epoch.

Motivated by these advances, we present xImagand-DKI, a novel multi-view SMILES/Protein-to-PK/DTI (SP2PKDTI) diffusion model. Conditioned on SMILES and protein embeddings, xImagand-DKI is capable of simultaneously generating 9 PK properties and 3 DTI values. Our key contributions are as follows:

- Proposes an end-to-end framework that unifies PK property prediction and DTI modeling into a single foundational model, advancing solutions to data overlap sparsity by generating high-quality synthetic drug discovery data.

- Introduces multi-view domain knowledge infusion (DKI) methods that incorporate protein knowledge from the Gene Ontology (GO) (Aleksander et al., 2023) and various molecular fingerprints.

- Demonstrates how end-to-end training method combined with multi-view domain knowledge integration can effectively address the challenge of data overlap sparsity, bridging the gap between PK and DTI datasets.

Notably, xImagand-DKI generates dense synthetic data that addresses the challenges posed by sparse and non-overlapping PK and DTI datasets. Using xImagand-DKI, researchers can generate large synthetic PK and DTI assay data across thousands of ligands, enabling the exploration of polypharmacy and drug combination research questions, at a fraction of the cost of conducting *in vitro* or *in vivo* PK assay panels.

## 2 BACKGROUND

Diffusion methods leverage families of probability distributions to model complex datasets in a way that enables computationally tractable learning, sampling, inference, and evaluation (Guo et al., 2023). DDPM (Jonathan et al., 2020) operates by first systematically destroying the structure in the data through a forward process, and then learning to reconstruct it from noise via a reverse generative process. Recent literature has highlighted significant advances in the use of diffusion models for small-molecule generation (Huang et al., 2023; Hoogeboom et al., 2022; Satorras et al., 2021; Vignac et al., 2023), conditional generation of drug PK properties (Hu et al., 2025; 2024), and multi-view fusion for DTI prediction (Ning et al., 2025; Wang et al., 2022; Suryanarayanan et al., 2025). In this study, we propose xImagand-DKI, a unified multi-view approach that leverages both molecular and protein perspectives for synthetic data generation in drug property and DTI prediction. Specifically, xImagand-DKI integrates molecular multi-views through circular and

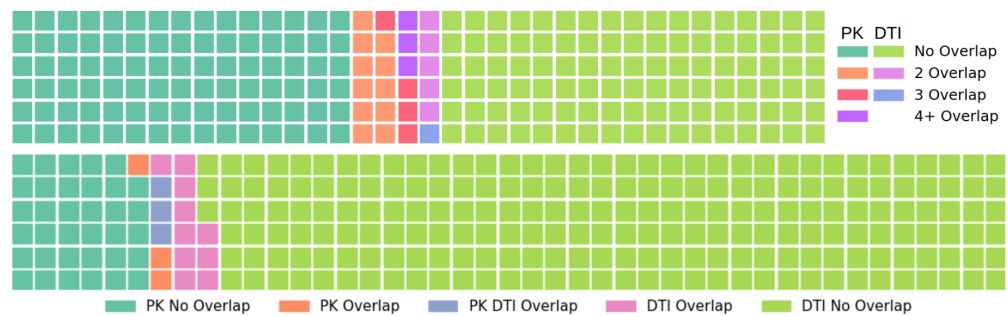

Figure 2: Visualizing data overlap sparsity between PK datasets and between DTI datasets (top), and between PK and DTI datasets (bottom). When compared to the total size of DTI and PK data points, 700k and 17k, respectively, we see data overlap sparsity, with a small percentage of molecules that belong to at least two datasets. We observe 16% of PK and 4.7% of DTI molecules with overlap.

| Dataset | Caco. | Lipo. | AqSo. | Free. | PPBR | VDss | Half | ClH | ClM | Total |
|---|---|---|---|---|---|---|---|---|---|---|
| DTI Overlap | 338 | 2789 | 1189 | 112 | 1241 | 184 | 486 | 698 | 794 | 4883 |
| PK Overlap | 179 | 1751 | 884 | 527 | 1296 | 163 | 337 | 879 | 1018 | 2772 |
| Dataset Size | 906 | 4200 | 9982 | 642 | 1797 | 1111 | 665 | 1020 | 1102 | 17k |

Table 1: Number of overlapping molecules for each 9 PK dataset with DTI and other PK datasets. We observe that there is a greater number of unique molecules in PK datasets that overlap with DTI datasets compared to other PK datasets.

structural molecular fingerprints, and protein multi-views using omics relationships derived from the Gene Ontology knowledge base.

## 2.1 Drug Discovery Data Overlap Sparsity

Drug discovery fails for two main reasons (Hughes et al., 2011): lack of efficacy and safety concerns. Understanding the relationship between solubility, toxicity, molecular structure, and drug response is essential for effective drug development (Kawabata et al., 2011; Bhalani et al., 2022), as these properties play a critical role in shaping a compound's absorption, distribution, metabolism, excretion, and Toxicity (ADMET) profile, as well as its therapeutic window and overall clinical viability. In this work, xImagand-DKI is trained to conditionally generate both drug properties and DTI data, addressing these challenges through a unified framework.

PK broadly describes what the body does to a drug, encompassing absorption (how the drug is taken up into the body), bioavailability (the extent to which the active drug enters systemic circulation), distribution (how the drug spreads through tissues), metabolism (how the body breaks down the drug), and excretion (how the drug is removed from the body). Issues related to PK properties are among the primary causes for compound attrition in small-molecule drug development (Kola, 2008), making accurate PK computational tools increasingly vital; and recent advances have significantly improved their capabilities (Waring et al., 2015; Davies et al., 2020; Ahmed et al., 2021).

DTI prediction examines the relationships between drugs and their biological targets, providing insights into the molecular pathophysiology of diseases (Askr et al., 2023; Kim et al., 2021). Accurately modelling DTI is crucial for applications such as drug repurposing, high-throughput screening, lead optimization, polypharmacy and drug combination research (Kim et al., 2021). Extending PK profiling across large arrays of ligands is often cost-prohibitive due to high expense of functional assays. An analysis of the overlap of 9 PK and 3 DTI datasets used in this study indicates that there is limited overlap between individual PK and DTI datasets, especially when considering overlaps of more than 2 datasets (Figure 2). Similarly, there is limited overlap between PK and DTI datasets, with only 0.7% of all DTI molecules having some PK value overlap. This fragmentation poses a

major barrier for researchers aiming to address complex questions that require integrated data, such as those in polypharmacy and drug combination studies.

## 2.2 DRUG DISCOVERY DOMAIN KNOWLEDGE

The Gene Ontology (GO) knowledge base is one of the most widely used resources in bioinformatics, offering structured annotations that describe the functions of genes and proteins across species. However, despite its biological richness, GO has rarely been directly integrated into deep learning models for drug discovery tasks. This underutilization stems partly from the dominance of sequence-based representations, which, although effective, often fail to capture the functional hierarchies and semantic relationships encoded in GO. Motivated by this limitation, we aim to enhance the quality of target protein embeddings by incorporating ontology-based information alongside sequence-level features.

Molecular fingerprints are bit strings that encode the structural information of a molecule, such as the presence or absence of specific chemical groups, atom types, or topological features (Hu et al., 2023). Molecular fingerprints offer a versatile representation where different algorithms tailored to capture different aspects of molecular structure, such as key-based fingerprints and hash fingerprints. Key-based fingerprints, including MACCS (Durant et al., 2002) and RDKit (Landrum, 2013), utilize a predefined fragment library to encode each molecule into a binary bit stream according to its substructure. Hash-based fingerprints such as Morgan fingerprints (Morgan, 1965) encode substructures in a molecule based on paths around atoms in a molecule. Leveraging fingerprints alongside SMILES representations in parallel increases the generalizability of models (Schimunek et al., 2023).

## 3 METHODOLOGY

xImagand-DKI is an SP2PKDTI diffusion model conditioned on learned SMILES and protein embeddings from their respective encoder models to generate target PK properties and DTI values. xImagand-DKI resembles a typical vision transformer architecture (Dosovitskiy et al., 2021); see Figure 1. 1D patches are computed from the classifier-free guidance of SMILES and protein embeddings and concatenated with PK class tokens. Diffusion step embeddings are generated using sinusoidal position encodings (Vaswani et al., 2023). Patches are then fed alongside sinusoidal step embeddings (Ho et al., 2021) to a transformer base. As the data is sparse over ligands, we apply masking when computing the loss to flow gradients from known PK and DTI values during training. Exponential Moving Average (EMA) (Tarvainen & Valpola, 2018) is applied to the base model during training to generate the final model used for sampling. Additional training details about pre-trained encoders and hyperparameters can be found in appendix A.1.

## 3.1 DIFFUSION MODEL

Given samples from a data distribution $q(x_0)$, we are interested in learning a model distribution $p_\theta(x_0)$ that approximates $q(x_0)$ and is easy to sample from. (Jonathan et al., 2020) considers the following Markov chain with Gaussian transitions parameterized by a decreasing sequence $\alpha_{1:T} \in (0, 1]^T$:

$$q(x_{1:T}|x_0) := \mathcal{N}(x_{1:T}|\sqrt{\alpha_{1:T}}x_0, (1 - \alpha_{1:T})\mathbf{I}) \tag{1}$$

This is called the *forward process*, whereas the latent variable model $p_\theta(x_{0:T})$ is the generative process, approximating the *reverse process* $q(x_{t-1}|x_t)$. The forward process of $x_t$ can be expressed as a linear combination of $x_0$ and noise variable $\epsilon$:

$$x_t = \sqrt{\alpha_t}x_0 + \sqrt{1 - \alpha_t}\epsilon \tag{2}$$

We train with the simplified objective:

$$L(\epsilon_\theta) := \sum_{t=1}^{T} \mathbb{E}_{x_0 \sim q(x_0), \epsilon_t}[||\epsilon_\theta^{(t)}(x_t) - \epsilon_t||_2^2] \tag{3}$$

where $\epsilon_\theta := \{\epsilon_\theta^{(t)}\}_{t=1}^{T}$ is a set of T functions, indexed by t, each with trainable parameters $\theta^{(t)}$.

## 3.2 Infusing Relationships from Gene Ontology

We leverage PO2Vec (Li et al., 2024), a recent embedding technique that transforms GO structures into continuous vector representations. Intuitively, PO2Vec relates the similarity between two terms $t_i$ and $t_j$ to the length of the shortest path between $t_i$ and $t_j$ in the GO. PO2Vec defines the shortest path based on three cases: (1) direct reachability $\mathcal{Q}_{dr}(t_i)$, if there exists a directed path starting at $t_i$ and ends at $t_j$; (2) indirect reachability $\mathcal{Q}_{ir}(t_i)$, if there exists a term $t_k$, reachable from both $t_i$ and $t_j$; (3) unreachable $\mathcal{Q}_{ur}(t_i)$, if $t_i$ and $t_j$ are neither directly or indirectly reachable from $t_i$.

PO2Vec applies contrastive learning to learn a partial order by sampling positive samples $t_i^+$ from $\mathcal{Q}_{dr}(t_i)$ or $\mathcal{Q}_{ir}(t_i)$ with specified shortest path length and k negative samples $\mathcal{N}(t_i)$ from indexed $\mathcal{Q}_{dr}(t_i)$, $\mathcal{Q}_{ir}(t_i)$, and $\mathcal{Q}_{ur}(t_i)$ with greater lengths. With $s(x, y)$ as cosine similarity between $x, y$, PO2Vec utilizes InfoNCE (van den Oord et al., 2019) defined by the following:

$$\mathcal{L}_{GO} = -\sum_{i=1}^{m} log \frac{s(t_i, t_i^+)}{\sum_{t_j \in \mathcal{N}(t_i) \cup \{t_i^+\}} s(t_i, t_j)} \tag{4}$$

The resulting GO term embeddings are then aggregated via average pooling over the annotated terms to obtain functional representations of genes. By integrating PO2Vec with ProtBert-derived sequence embeddings prior to the diffusion process, our model benefits from both molecular sequence information and ontology-driven semantics, leading to more biologically meaningful target representations.

## 3.3 Infusing Structural and Circular Drug Fingerprints

We leverage FPFormer, a novel embedding model pre-trained on both structural and circular fingerprints from ChemBL (Gaulton et al., 2017) and Moses (Polykovskiy et al., 2020). FPFormer utilizes a novel tokenization methodology that converts different sparse fingerprints into a chemical language and sequence, compatible with masked language modelling pre-training and embedding techniques. Molecular fingerprints can be computed from SMILES strings, where each methods looks to represent and encode different aspect of a molecule Cereto-Massagué et al. (2015). We utilize a mixture of structural, circular, and atom-pair fingerprints ECFP4, FCFP6, MACCS, AVALON, TOPTOR, and ATOMPAIR to pre-train our FPFormer model to generate meaningful molecular embedding representations, complementing learned SMILES embeddings.

### 3.3.1 Pre-trained SMILES and Protein Encoders

SP2PKDTI diffusion models need powerful semantic SMILE and protein encoders to capture the complexity of arbitrary chemical and biological structure inputs. Given the sparsity and small size of PK datasets, encoders trained on specific SMILES-Pharmacokinetic or SMILES-Protein pairs are infeasible (Huang et al., 2021). Many transformer-based foundational models such as ChemBERTa (Chithrananda et al., 2020; Ahmad et al., 2022), SMILES-BERT (Wang et al., 2019), and MOLGPT (Bagal et al., 2021) have been pre-trained to deeply understand molecular and chemical structures and properties. Similar transformer-based foundation models such as ProtBERT (Elnaggar et al., 2020) have been pre-trained to deeply understand protein structures and properties. After pre-training, these foundational models can then be fine-tuned for various downstream molecular and protein tasks.

We test SMILES embeddings from ChemBERTa (Ahmad et al., 2022) and protein embeddings from ProtBERT (Elnaggar et al., 2020), trained on SMILES-only and protein-only corpora, respectively. Both embedding models were collected through the Huggingface (Wolf et al., 2020) Model Hub. Similar to (Saharia et al., 2022), we freeze the weights of our embedding models. Because embeddings are computed offline, freezing the weights minimizes computation and memory footprint for embeddings during model training.

## 4 Experiments

In the following, we describe the model training details and compare our synthetic data to real data, in terms of machine learning efficiency (MLE) and univariate and bivariate statistical distributions. We then discuss ablation studies and key findings. The metrics for MLE, univariate, and bivariate

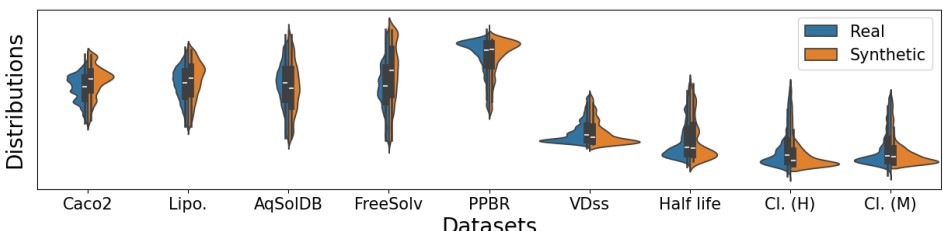

Figure 3: Distributions of ligand PK properties. Blue, synthetic distributions; orange, real distributions.

evaluations are further defined in their respective subsections. We compare xImagand-DK with baselines of Conditional GAN (cGAN) (Mirza & Osindero, 2014) and Syngand (Hu et al., 2024). SMILES-embeddings from a pre-trained T5 model are used conditionally by the cGAN model to generate PK properties as output for a specific drug. Additional baseline and DKI ablation results are provided in appendix A.2.

### 4.1 PHARMACOKINETIC AND DRUG-TARGET INTERACTION DATASETS

All 9 PK and 3 DTI datasets are collected from TDCommons (Huang et al., 2021). We select PK datasets suitable for regression from the ADMET categories. We select DTI datasets from BindingDB (Liu et al., 2007) covering properties such as inhibition constant ($K_i$), dissociation constant ($K_d$), and half maximal inhibitory concentration (IC50). Revealing the overlap sparsity between DTI and PK, out of around 700k molecules from BindingDB, only around 5k molecules (0.7%) have PK properties defined from one of the 9 PK datasets.

The **inhibition constant** is a measure of how strongly an inhibitor binds to an enzyme, effectively indicating the inhibitor's potency. BindingDB has 375k pairs of $K_i$ values from 175k drugs and 3k proteins. The **dissociation constant** quantifies binding affinity between a drug and its target protein, defined as the free ligand concentration at which 50% of the protein binding sites are occupied at equilibrium. BindingDB has 52k pairs of $K_d$ values from 11k drugs and 1.5k proteins. The **half maximal inhibitory concentration** is a measure of the potency of a substance in inhibiting a specific biological or biochemical function. BindingDB has 991k pairs of IC50 values from 550k drugs and 5k proteins.

**Caco-2** (Wang et al., 2016) is an absorption dataset containing rates of 906 drugs passing through the Caco-2 cells, approximating the rate at which the drugs permeate through the human intestinal tissue. **Lipophilicity** (Wu et al., 2018) is an absorption dataset that measures the ability of 4,200 drugs to dissolve in a lipid (e.g. fats, oils) environment. **AqSolDB** (Sorkun et al., 2019) is an absorption dataset that measures the ability of 9,982 drugs to dissolve in water. **FreeSolv** (Mobley & Guthrie, 2014) is an absorption dataset that measures the experimental and calculated hydration-free energy of 642 drugs in water.

**Plasma Protein Binding Rate (PPBR)** (Wenlock & Tomkinson, 2016) is a distribution dataset of percentages for 1,614 drugs on how they bind to plasma proteins in the blood. **Volume of Distribution at steady state (VDss)** (Lombardo & Jing, 2016) is a distribution dataset that measures the degree of concentration for 1,130 drugs in body tissue compared to their concentration in blood.

**Half Life** (Obach et al., 2008) is an excretion dataset for 667 drugs on the duration for the concentration of the drug in the body to be reduced by half. **Clearance** (Di et al., 2012) is an excretion dataset for around 1,050 drugs on two clearance experiment types, microsome and hepatocyte. Drug clearance is defined as the volume of plasma cleared of a drug over a specified time (Huang et al., 2021). **Acute Toxicity (LD50)** (Zhu et al., 2009) is a toxicity dataset that measures the most conservative dose for 7,385 drugs that can lead to lethal adverse effects.

#### 4.1.1 DATA PROCESSING

We first merge all 9 PK datasets to create a unified dataset containing 17k drugs over 9 unique PK columns for training and testing (90%/10% split) our models. We merge all data from 3 DTI datasets

| Model | PKs | | | | | | | | | DTIs | | |
|-------|-----|-----|-----|-----|-----|-----|-----|-----|-----|-----|-----|-----|
| | C2 | Li. | Aq | FS | PP | VD | HL | ClH | ClM | $K_d$ | $K_i$ | I50 |
| Sygd | 0.62 | 0.53 | 0.34 | 0.50 | 0.66 | 0.81 | 0.85 | 0.59 | 0.58 | ∅ | ∅ | ∅ |
| cGAN | 0.19 | 0.16 | 0.17 | 0.18 | 0.25 | 0.24 | 0.28 | 0.32 | 0.29 | 0.32 | 0.08 | 0.13 |
| Imgd | 0.19 | 0.12 | 0.13 | 0.18 | 0.20 | 0.27 | 0.36 | 0.20 | 0.19 | 0.27 | 0.13 | 0.11 |
| No DKI | **0.12** | 0.08 | 0.07 | 0.13 | 0.11 | 0.12 | 0.15 | **0.13** | 0.18 | 0.26 | 0.07 | 0.09 |
| **Ours** | 0.13 | **0.07** | **0.07** | **0.12** | **0.09** | **0.08** | **0.15** | 0.15 | **0.15** | **0.24** | **0.06** | **0.07** |

Table 2: Average Hellinger distance across 30 generated synthetic target property datasets for ablation experiment configurations. The best HD values for each ablation test are bolded. We compare our proposed model with and without DKI to existing benchmarks of Imagand, Syngand, and cGAN.

to create a unified dataset containing 1.2M drug-protein pairs with 3 dti columns for training and testing (90%/10% split) for our models. Data from 3 DTI datasets are log-transformed.

We apply a Gaussian Quantile Transform to both PK and DTI datasets before min-max scaling between the range of $[-1, 1]$. After removing outliers ($Q1 - 1.5IQR$ lower and $Q3 + 1.5IQR$ upper bound), we are left with 16.5k drugs from the original 17K drugs and 1.1M pairs from 1.2M DTI pairs. Outliers are removed to ensure that Min-Max normalization does not cause unwarranted skewness in our trainset distribution, causing issues for model training. Before infilling null values using inverse transform sampling, we store the null masks for each drug for the masked loss function.

## 4.2 UNIVARIATE COMPARISONS TO REAL DATA

Hellinger distance (HD) quantifies the similarity between two probability distributions and can be used as a summary statistic of differences for each PK target property between real and synthetic datasets. Given two discrete probability distributions $P = \{p_1, p_2, ..., p_n\}$ and $Q = \{q_1, q_2, ..., q_n\}$, the HD between $P$ and $Q$ is expressed in Equation 5.

$$HD^2(p,q) = \frac{1}{2} \sum_{i=1}^{n} \left(\sqrt{p_i} - \sqrt{q_i}\right)^2 \tag{5}$$

With scores ranging between 0 to 1, HD values closer to 0 indicate smaller differences between real and synthetic data and are thus desirable.

Figure 3 shows the distributions of PK synthetic data generated by xImagand-DKI with the real data. Computing the Hellinger distance, Table 2, we see an average of 0.11, meaning that our model produces synthetic data that closely resembles the distribution of real data. Additional DKI HDs ablations are in appendix A.2.1. Table 2 shows that data generated from our proposed architecture more closely resembles real data compared to other models.

## 4.3 BIVARIATE CORRELATIONS OF SYNTHETIC DATA

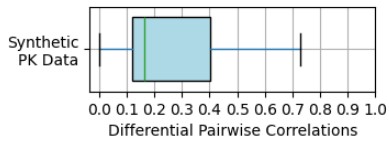

Figure 5: Boxplot of Pairwise correlations

In addition to univariate comparisons, synthetic PK target properties can be compared to real data in terms of bivariate pairwise distributions and correlations. Differential Pairwise Correlations (DPC) provides a multivariate metric for evaluating the quality of synthetic data when compared to real data. We define the DPC as the absolute difference between the bivariate correlation coefficient of real and synthetic data, denoted by subscripts $r$ and $s$, respectively, as shown in Equation 6.

$$\Delta CV_{cont_{XY}} = |\rho_{XY_r} - \rho_{XY_s}| \tag{6}$$

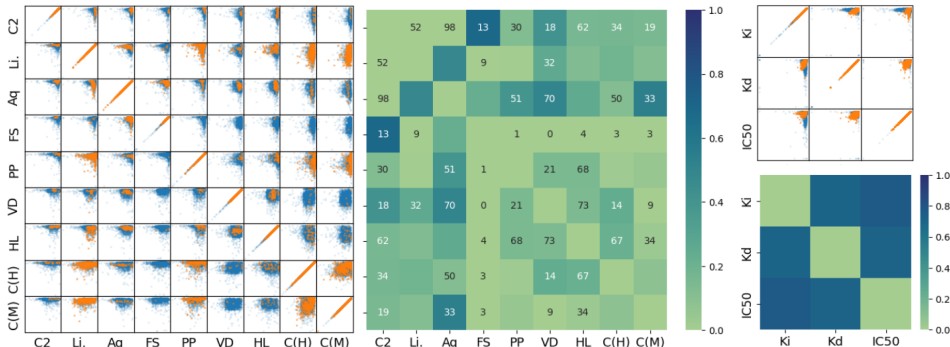

Figure 4: Overview of bivariate comparison between synthetic and real data. We show pairwise scatter plots for pairs of PK and DTI target properties. Real data is marked in orange, and synthetic data is marked in blue. The heatmap plots are the Differential Pairwise Correlations (DPC) using Pearson Correlation Coefficient for pairs of PK target properties between real and synthetic data.

| | | Models | | | | | | Models | | | |
|---|---|---|---|---|---|---|---|---|---|---|---|
| | | Real | cGAN | Imgd | **Ours** | | | Real | cGAN | Imgd | **Ours** |
| C2 | mse | 0.63 | 0.17 | 0.13 | **0.06** | HL | mse | 0.53 | 0.28 | 0.26 | **0.07** |
| | R2 | -3.2 | -0.08 | **0.14** | -0.13 | | R2 | -1.6 | -0.54 | -0.28 | **-0.09** |
| | pcc | 0.35 | 0.34 | **0.43** | 0.35 | | pcc | 0.16 | 0.13 | 0.03 | **0.17** |
| Li. | mse | 0.17 | 0.14 | 0.15 | **0.09** | CH | mse | 1.9 | 0.43 | 0.43 | **0.15** |
| | R2 | 0.04 | **0.19** | 0.14 | 0.01 | | R2 | -4.2 | -0.15 | -0.20 | **-0.13** |
| | pcc | 0.50 | 0.47 | 0.41 | **0.49** | | pcc | 0.11 | **0.14** | 0.10 | 0.10 |
| Aq | mse | 0.075 | 0.07 | 0.08 | **0.07** | CM | mse | 0.72 | 0.20 | 0.21 | **0.04** |
| | R2 | 0.56 | **0.57** | 0.53 | 0.38 | | R2 | -2.6 | **-0.04** | -0.04 | -0.06 |
| | pcc | 0.76 | **0.76** | 0.73 | 0.75 | | pcc | 0.13 | 0.25 | **0.25** | 0.17 |
| FS | mse | 0.62 | 0.20 | 0.17 | **0.11** | $K_d$ | mse | 0.11 | 0.11 | 0.11 | 0.11 |
| | R2 | -2.5 | -0.09 | **0.08** | -0.22 | | R2 | 0.22 | 0.23 | 0.23 | **0.23** |
| | pcc | 0.38 | **0.42** | 0.39 | 0.39 | | pcc | 0.50 | 0.49 | 0.50 | **0.50** |
| PP | mse | 3.5 | 0.26 | 0.26 | **0.04** | $K_i$ | mse | 0.11 | 0.11 | 0.11 | 0.11 |
| | R2 | -13 | -0.08 | -0.06 | **-0.05** | | R2 | 0.21 | 0.21 | 0.22 | **0.22** |
| | pcc | 0.10 | **0.23** | 0.22 | 0.10 | | pcc | 0.46 | 0.46 | 0.47 | **0.47** |
| VD | mse | 0.54 | 0.21 | 0.20 | **0.04** | I50 | mse | 0.13 | 0.13 | 0.13 | 0.13 |
| | R2 | -1.8 | -0.06 | **-0.02** | -0.07 | | R2 | 0.16 | 0.16 | 0.16 | **0.16** |
| | pcc | 0.23 | **0.31** | 0.30 | 0.21 | | pcc | 0.40 | 0.40 | 0.40 | 0.40 |

Table 3: Comparing drug discovery Machine Learning Efficiency (MLE) regression performances between different models and with real train data. Mean Squared Error (mse), R-Squared (R2), and Pearson Correlation Coefficient (pcc) values are averaged over 30 trials, with the best scores on the real testset bolded. R2 and pcc values are scale-adjusted relative to Real-Real with cGAN and Imagand results.

where $X$ and $Y$ denote the two continuous variables, whereas $\rho_{XY}$ is the correlation coefficient for $X$ and $Y$. If the real and synthetic PK target property datasets are highly similar (i.e., the synthetic dataset closely resembles the real dataset), then the absolute difference would be close to 0 or very small, as seen in Figure 5. Heatmaps in Figure 4 show DPC on the Pearson correlation coefficient (pcc) between both PK and DTI data points. These results indicate that the generated synthetic PK target properties resemble real data in pairwise correlations.

## 4.4 Performance on Real-World Tasks

Machine Learning Efficiency (MLE) is a measure that assesses the ability of the synthetic data to replicate a specific use case (Dankar & Ibrahim, 2021; Basri et al., 2023; Borisov et al., 2022). MLE represents the ability of the synthetic data to replace or augment real data in downstream use cases. To measure MLE, two models are trained separately, one with synthetic and the other with real data. Then their performance is compared using Mean-Squared Error (mse), R-Squared (R2), and Pearson Correlation Coefficient (pcc), is evaluated on real data test sets. Further details on our MLE experiment setup are included in appendix A.3.

Table 3 shows the results of the PK and DTI regression tasks using real and synthetic augmented datasets. Results of these experiments suggest that a synthetic augmented dataset can outperform real data with statistical significance over many PK datasets. Additional DKI MLE ablations are in appendix A.2.1. Additional tasks will be explored in future work as well as improving MLE performance for $K_i$ and $IC50$ DTI tasks. We see that synthetic data from both cGAN and xImagand-DK can improve MLE over using only the real data.

## 5 Limitations and Future Work

Our work is a major step towards building a new class of foundational models for drug discovery trained over a diverse range of datasets. Given the problem of data overlap sparsity, xImagand-DKI can be utilized primarily as a *in silico* pre-clinical tool, aimed to reduce the costs of *in vitro* experiments and high-throughput screening. As a research tool, scientists can utilize our models to investigate and generate properties for novel molecules to be used for downstream PBPK simulations without costly assays. Even as an initial step, xImagand-DKI has many real-world pre-clinical applications where data overlap sparsity and data scarcity are challenges.

- **Limited applicability to *in vivo* applications.** Although we cover a wide variety of ADMET and DTI datasets, most of these datasets are *in vitro*. *In vivo* experiments provides real-world data that complements *in vitro* studies, where that data can be used to further improve the performance of our models.

- **Applicability only to numerical drug discovery datasets.** With the limitations of our diffusion methods, we are restricted to utilizing only numerical datasets. This limits the types of datasets that our model is applicable with, such as ToxCast (Richard et al., 2016) classification datasets of over 600 experiments.

- **Extending beyond PK/DTI drug discovery tasks.** PK/DTI data and research makes up only a small section of pre-clinical drug-discovery. AI for lead optimization, de novo drug design, and protein-docking are other interconnected research innovating pre-clinical drug discovery.

Future work will look to extend our model to *in vivo* datasets and investigate how our generated data can be used for quantitative in vitro-to-in vivo extrapolation. We will look to extend our model to categorical diffusion methods as well as investigating integration with other drug discovery tasks in lead optimization, de novo generation, and protein-docking. Extending our model to categorical datasets and other drug discovery tasks will allow us to benchmark and train our model on additional drug discovery datasets, adaptable for a larger number of tasks.

## 6 Conclusions

The SMILES/Protein to PK/DTI model xImagand-DTI generates synthetic PK and DTI target property data that closely resembles real data in univariate and for downstream tasks. xImagand-DKI provides a solution for the challenge of sparse overlapping PK and DTI target property data, allowing researchers to generate data to tackle complex research questions and for high-throughput screening. Future work will expand xImagand-DKI to categorical PK and DTI properties, and scale to more datasets and larger model sizes. For future work, we will look to extend our model to include *in vivo* datasets and to investigate new applications of xImagand-DKI for QIVIVE.

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

# A APPENDIX

## A.1 TRAINING DETAILS

| xImagand-DK Model | | Diffusion Training | |
|---|---|---|---|
| Layers | 12 | Learning Rate | 1e-3 |
| Heads | 16 | Weight Decay | 5e-2 |
| MLP Dim. | 768 | Epoch | 3000 |
| Emb. Dropout | 10% | Batch Size | 256 |
| Num Patches | 48 | Warmup | 200 |
| Drug Emb. Size | 768 | Timesteps (Train) | 2000 |
| Time Emb. Size | 64 | Timesteps (Infer.) | 150 |
| PK Emb. Size | 256 | EMA Gamma ($\gamma$) | 0.994 |
| Prot Emb. Size | 1024 | | |
| Drug DKI Emb. | 768 | | |
| Prot DKI Emb. | 256 | | |

Table 4: List of xImagand-DK Model Hyperparameters used across experiments. Model hyperparameters include the number of layers, heads, multilayered perceptron (MLP) size, embedding dropout, and sizes for the conditional, time, and pharmacokinetic (Y) embeddings. Training hyperparameters include the learning rate, weight decay, number of epochs, batch size, warmup, diffusion timesteps used for training and inference, and the Exponential Moving Average (EMA) Gamma ($\gamma$).

|          | DTIs |      |      |
|----------|------|------|------|
| **Model** | **$K_d$** | **$K_i$** | **I50** |
| cGAN     | 0.32 | 0.08 | 0.13 |
| Imgd     | 0.27 | 0.13 | 0.11 |
| No DKI   | 0.26 | 0.07 | 0.09 |
| FP DKI   | 0.28 | 0.08 | 0.09 |
| FP-GO DKI | **0.23** | 0.11 | 0.11 |
| **GO DKI** | 0.24 | **0.06** | **0.07** |

Table 5: Hellinger distance across datasets for ablation experiment configurations.

We train a 19M parameter model for S2PK synthesis. Model hyperparameters were not optimized and are described in Table 4. For classifier-free guidance, we joint-train unconditionally via dropout zeroing out sections of the SMILES embeddings with 10% probability for all of our models. For the machine learning efficiency, and univariate and bivariate distribution analysis, we utilize DeBERTa embeddings trained on PubChem and DLGN for infilling and as the noise model. We compare our model configuration to other possible configurations in the ablation experiments. All experiments were conducted using a single NVIDIA GeForce RTX 3090 GPU.

### A.1.1 STATIC THRESHOLDING

We apply elementwise clipping the PK predictions to $[-1, 1]$ as static thresholding, similar to (Saharia et al., 2022; Jonathan et al., 2020). Since PK data is min-max scaled to the same $[-1, 1]$ range as a preprocessing step, static thresholding is essential to prevent the generation of invalid and out-of-range PK values.

### A.1.2 CLASSIFIER-FREE GUIDANCE

Classifier guidance uses gradients from a pre-trained model to improve quality while reducing diversity in conditional diffusion models during sampling (Dhariwal & Nichol, 2021). Classifier-free guidance (Ho & Salimans, 2022) is an alternative technique that avoids this pre-trained model by jointly training a diffusion model on conditional and unconditional objectives via dropping the condition (i.e. with 10% probability). We condition all diffusion models on learned SMILES embedding and sinusoidal time embeddings using classifier-free guidance through dropout (Ho & Salimans, 2022; Srivastava et al., 2014).

### A.2 ABLATION STUDIES

### A.2.1 DTI DOMAIN KNOWLEDGE INFUSION

We conduct ablations for drug-target interaction domain knowledge infusion. From Table 5, we find that domain knowledge infusion with the human GO embeddings increases the quality of generated synthetic data over other DKI techniques. Similarly, computing MLE across DTI DKI ablations, Table 6 shows that human GO embeddings improves the MLE performance of the synthetic data on downstream tasks. There is limited gain in MLE performance across $K_d$ and $IC50$ given all ablation experiments, future work will look to investigate this phenomenon.

### A.2.2 ADDITIONAL BASELINES

We compare xImagand-DK with a baseline in Conditional GAN (cGAN) (Mirza & Osindero, 2014) with 1.8M parameters and Syngand (Hu et al., 2024) with 9M parameters. SMILES-embeddings from a pre-trained T5 model are used conditionally by the cGAN model to generate PK properties as output for a specific drug. Compared to earlier results, Table 7, Figure 7, and Figure 6 shows that xImagand-DK is able to generate more realistic synthetic data compared to cGAN and Syngand.

|  |  | | DKI Model Ablations | | | |
|  |  | Real | None | FP | FP-GO | **GO** |
|---|---|---|---|---|---|---|
| | mse | 0.11 | 0.11 | 0.10 | 0.10 | 0.10 |
| $K_d$ | R2 | 0.22 | 0.23 | 0.26 | 0.25 | **0.26** |
| | pcc | 0.50 | 0.50 | 0.50 | 0.50 | **0.51** |
| | mse | 0.11 | 0.11 | 0.11 | 0.11 | 0.11 |
| $K_i$ | R2 | 0.21 | 0.22 | 0.21 | 0.21 | **0.22** |
| | pcc | 0.46 | 0.47 | 0.46 | 0.46 | **0.47** |
| | mse | 0.13 | 0.13 | 0.13 | 0.13 | 0.13 |
| I50 | R2 | 0.16 | 0.16 | 0.16 | 0.16 | **0.16** |
| | pcc | 0.40 | 0.40 | 0.40 | 0.40 | 0.40 |

Table 6: Comparing drug discovery Machine Learning Efficiency (MLE) regression performances between different ablation models and with real train data. Mean Squared Error (mse), R-Squared (R2), and Pearson Correlation Coefficient (pcc) values are averaged over 30 trials, with the best scores on the real testset bolded.

| | Mean | | | | | Std | | | | |
|---|---|---|---|---|---|---|---|---|---|---|
| Data | Real | Ours | Imgd | cGAN | Sygd | Real | Ours | Imgd | cGAN | Sygd |
| Caco2 | 0.12 | 0.13 | 0.14 | 0.14 | 0.58 | 0.39 | 0.39 | 0.38 | 0.27 | 0.12 |
| Lipo | 0.18 | 0.19 | 0.18 | 0.20 | 0.62 | 0.42 | 0.41 | 0.39 | 0.30 | 0.19 |
| AqSol | 0.11 | 0.18 | 0.11 | 0.13 | 0.10 | 0.41 | 0.39 | 0.36 | 0.29 | 0.18 |
| FreeSolv | 0.10 | 0.17 | 0.12 | 0.10 | 0.27 | 0.42 | 0.42 | 0.40 | 0.29 | 0.13 |
| PPBR | 0.57 | 0.59 | 0.56 | 0.63 | 0.97 | 0.50 | 0.49 | 0.47 | 0.37 | 0.09 |
| VDss | -0.60 | -0.61 | -0.62 | -0.66 | -0.98 | 0.44 | 0.43 | 0.39 | 0.31 | 0.06 |
| Halflife | -0.56 | -0.60 | -0.56 | -0.61 | -0.98 | 0.45 | 0.43 | 0.42 | 0.32 | 0.06 |
| ClH | -0.55 | -0.58 | -0.56 | -0.59 | -0.97 | 0.61 | 0.56 | 0.55 | 0.47 | 0.09 |
| ClM | -0.67 | -0.69 | -0.68 | -0.74 | -0.98 | 0.45 | 0.44 | 0.38 | 0.31 | 0.06 |

Table 7: Comparing mean and standard deviation values between real and synthetic target property values, rounded to two significant figures.

## A.3 MLE EXPERIMENT SETUP

For this experiment, we train Linear Regression (LR) models using T5 chemical and ProtBERT embeddings to predict each PK and DTI target property value. To ensure an adequately sized test set (>300 ligands, i.e. >10% size of our synthetic data) to evaluate our downstream models, we divide real data into segments $A_r$ and $B_r$ using a 50%/50% split. To ensure a synthetic test set similar in size to real data test sets ($\sim$ 300 ligands), we divide synthetic data into segments $A_s$ and $B_s$ using a 90%/10% split. The real train set is defined as $A_r$, and the real test set is defined as $B_r$. The augmented train set is defined as $A_r \cup A_s$, and the augmented test set is defined as $B_r \cup B_s$. Outliers are removed from both real and augmented train and test sets based on $Q1 - 1.5\text{IQR}$ lower and $Q3 + 1.5\text{IQR}$ upper bounds on the synthetic data.

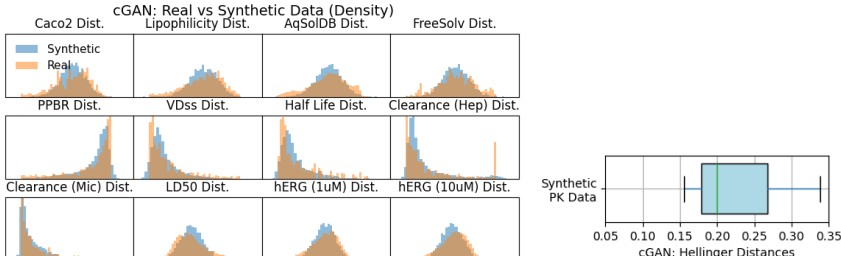

Figure 6: Distributions of ligand PK properties and synthetic PK Data Hellinger Distances (HDs) for cGAN. Blue, synthetic distributions; orange, real distributions.

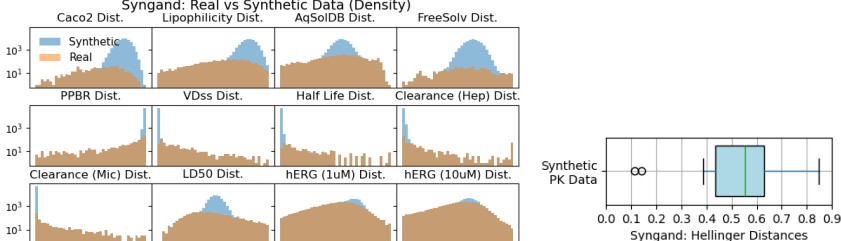

Figure 7: Distributions of ligand PK properties (log-scale) and synthetic PK Data Hellinger Distances (HDs) for Syngand. Blue, synthetic distributions; orange, real distributions.