# OpenReview forum: "Domain Knowledge Infused Conditional Generative Models for Accelerating Drug Discovery"
_ICLR.cc/2026/Conference — Submitted to ICLR 2026_

### Official Review · Reviewer_HqWx · 2025-10-16

**Soundness:** 1
**Presentation:** 1
**Contribution:** 2
**Rating:** 0
**Confidence:** 3

**Summary:**

The paper presents xImagand-DKI, a conditional diffusion model designed to generate pharmacokinetic (PK) and drug–target interaction (DTI) data conditioned on SMILES and protein sequences. It addresses the issue of data overlap sparsity between PK and DTI datasets by integrating additional molecular and genomic knowledge from Gene Ontology and molecular fingerprints. The model aims to synthesize realistic PK data distributions and bridge missing links between heterogeneous drug discovery datasets. The approach is proposed as a means to support downstream tasks such as high-throughput screening and drug combination studies.
I think it is a good effort, but overall it lacks soundness, presentation clarity, motivation and good experimental depth for ICLR. There are many questionable approaches in these parts, which require more exploration of related works, experiments and analyses. The presentation, sequence of contents and clarity of the work also should be improved.

**Strengths:**

1. Tries to define and address the problem of data overlap sparsity in PK and DTI datasets.
2. Applies a suitable conditional diffusion model for multimodal data generation.
3. Integrates domain knowledge from Gene Ontology and molecular fingerprints.
4. Enables realistic PK data synthesis to support downstream drug discovery tasks.

**Weaknesses:**

1. In terms of novelty, there is no theoretical/technical novelty in this work, as all components have been used in drug discovery for quite some time. However, the application motivation is new, considering the presented literature.
2. The motivation of this work seems quite weak; *"SMILES and amino acid sequences, alone are likely not sufficient in fully capturing the complexity of natural entities,"* (L049) so we need fusion. Generally, data scaling is always believed to be helpful and increase performance, so nothing uncommon; but exploring how and why pharmacokinetic (PK) properties can help here in the model (from a biomedical perspective) can bring interesting insights for wet-lab researchers. Also, this effect should be studied in the proposed model framework too: how these PK properties are affecting the generation.
3. In L154-164, it is not clear how DTI and PK properties are connected, how and why the sparsity issue is important, and what problem it'll solve specifically.
4. Use of biological terms and related derivatives is poorly introduced. In DTI, ligands are the targets (T)? There are some more confusing terms like this.
5. No reference is available in Section 2.2 DRUG DISCOVERY DOMAIN KNOWLEDGE. Also, I need more details on GO: what information does it contain, what is the form of that information (tabular, embeddings, or what), and how is it used here (any preprocessing or such)?
6. The methodology requires more clarification:
    - In both diffusion models, there seem to be 2 inputs, so are they concatenated?
    - Rather than a general definition of the diffusion model, the mathematical details should be done according to the adaptation in this work. For example, start with SMILES ($s_i$) and ligands ($l_i$) and show everything in an end-to-end manner.
    - Lines 244-247 say that the authors pre-trained their FPFormer model using different fingerprints. But no details on why and how are available. Fingerprints like ECFP4 and MACCS are very capable of producing high-quality SMILES embeddings, as shown in many prior works. Is there any motivation for choosing FPFormer or any comparative study done to justify the use?
    - The difference between ChemBERTa and FPFormer, in terms of biological insights, is unclear. Both seem to provide exactly the same information: molecular and chemical structures and properties.
7. There is no ablation study or related justifications for data preprocessing mentioned in lines 340-342.
8. From table 2, it is noticeable that the improvements with DKI are incremental and minimal. There is no statistical justification that DKI helps or not. Also, another part is complexity; if the improvement is so minimal, is all the complexity worth it?
9. Section 4.4 lacks details on what the "real world" tasks are and why they are meaningful in terms of the problem and solution. Are they benefited by domain knowledge infusion? If yes, how? If not, why?
10. Table 3 shows multiple negative R-squared (R²) values, which seems like a negative sign. Statistically, this indicates that the model is not explaining any of the variance in the data, and the predictions are less accurate than if you just used the average value. There is no discussion on why this is happening and if this is acceptable or not (and why).
11. No discussion on how the results justify the motivation and solve the research gap. Apart from results and stats, the paper doesn't note how these can lead to meaningful pharmacological/biological applications, as stated in the motivation.

Minor Issues:
1. Lines 258/259 say ChemBERTa, but figure 1 says deBERTa. Needs clarification.

**Questions:**

1. The technical setup feels quite standard. What’s truly new here beyond applying existing models to this dataset? The motivation mentions the limitation of SMILES and sequences, but could you expand on what specific biological information is missing and how your added PK features fill that gap?
2. You mention that fusion of modalities helps. Can you give a biomedical rationale for why PK properties in particular might complement molecular and sequence data? That’d make the motivation stronger.
3. The link between DTI and PK properties in L154–164 is not clear. Are you saying that PK data helps model the binding behavior or just improves general representation?  Also, could you explain why sparsity matters here? What’s sparse (data, features, interactions?), and how does that affect performance?
4. There’s some confusion in terminology. For instance, in DTI, ligands are the *drugs* and targets are the *proteins*, right? But the text sometimes reads like the opposite.  Maybe define each biological term the first time it appears, especially ligand, receptor, target, binding affinity, etc. to help readers outside biochemistry.
5. Section 2.2 (Domain Knowledge) doesn’t have any references, can you cite some sources for how GO or other biological annotations are used in computational models? Also, how exactly is GO data used here? Is it a table, an embedding, or ontology-based graph features? Is there any preprocessing or encoding step for GO before feeding it into the model?
6. In the diffusion model, are both inputs concatenated or processed separately before fusion? The diffusion formulation feels generic. Could you rederive or summarize the key equations in your own context (starting from SMILES and ligand inputs)?
7. Why did you choose FPFormer for fingerprint embeddings instead of standard ones like ECFP4 or MACCS? Any empirical or conceptual reason?
8. What’s the real difference between ChemBERTa and FPFormer in terms of what biological or chemical features they encode? They sound overlapping.
9. There’s no ablation or sensitivity analysis for preprocessing choices (L340–342). It would be good to know if those steps really matter. From Table 2, the DKI improvement seems very minimal. Have you checked if the difference is statistically significant? If not, the added complexity might not be justified.
10. In Real-world relevance (Section 4.4), what exactly are the “real-world tasks”? Are these downstream biological prediction tasks or actual drug discovery steps?
11. In Table 3, several models show negative R². That means they perform worse than predicting the mean. Why is that happening?
12. The results show small gains, but it’s hard to connect them to the claimed biological motivation. Could you elaborate on how this helps pharmacological insight or drug design in practice? How do the results close the loop with the original problem statement? What did we learn biologically or methodologically beyond just getting a slightly better score?
13. No discussion on how the results justify the motivation and solve the research gap. Apart from results and stats, the paper doesn't note how these can lead to meaningful pharmacological/biological applications, as stated in the motivation.
14. Was there any comparative study done before choosing ChemBERTa and ProtBERT models for the task?
15. Was Hellinger distance (HD) used in any prior DTI or such studies (L348-352)? Is it an established metric for evaluating biological sequence generation?
16. In Figure 3 and Section 4.2, perform a statistical significance test to verify the similarity.
17. Mention the short forms (C2 Li. Aq FS PP VD HL ClH ClM) while introducing the terms at Section 4.1. Currently it is very hard to head and connect.

---

### Official Review · Reviewer_WycR · 2025-10-19

**Soundness:** 2
**Presentation:** 3
**Contribution:** 2
**Rating:** 4
**Confidence:** 3

**Summary:**

The paper proposes **xImagand-DKI**, a conditional diffusion model for drug discovery that jointly addresses **pharmacokinetics (PK)** and **drug–target interaction (DTI)** tasks. A key challenge motivating this work is **data overlap sparsity**, where PK and DTI datasets often have very limited molecule overlap, making it difficult to integrate information across assays. To address this, the model generates **synthetic PK and DTI data** conditioned on molecular SMILES and protein sequences. Crucially, it incorporates **domain knowledge infusion**: Gene Ontology–based embeddings for proteins and molecular fingerprints for SMILES, enhancing biological and chemical contextualization.

Experiments show that synthetic data from xImagand-DKI closely match real data distributions (evaluated with Hellinger distance and correlation metrics) and improve downstream machine learning efficiency when augmenting real datasets. Comparisons are made against baselines (cGAN and Syngand) and ablations of the domain knowledge components, with xImagand-DKI consistently outperforming them. While the approach demonstrates potential to mitigate data overlap sparsity and reduce reliance on expensive in vitro assays, it is currently limited to in vitro numerical datasets and does not isolate the benefits of jointly modeling PK and DTI. The authors suggest future work on extending to in vivo, categorical, and broader drug discovery tasks.

**Strengths:**

1. **Essential domain of drug discovery**: The paper tackles a highly relevant and impactful problem in computational drug discovery—data overlap sparsity between pharmacokinetics (PK) and drug–target interaction (DTI) datasets. This is a fundamental barrier in integrating experimental assays, and addressing it through generative modeling has the potential to reduce costs and accelerate preclinical research. The contribution is well-motivated in the context of the multi-billion-dollar cost and high failure rate of drug.
2. **Clear presentation of methodology**: The paper presents the proposed xImagand-DKI model with a structured and accessible explanation. The role of the diffusion backbone, the infusion of molecular and protein domain knowledge, and the experimental design are all clearly described, making the technical contributions understandable to both ML and drug discovery audiences. Figures and tables (e.g., distribution matching, MLE performance) are well-integrated to support the narrative.
3. **Abundant discussion and reflection**: The authors provide extensive discussion of their results, including distributional comparisons, downstream ML efficiency, and domain knowledge ablations. They also openly acknowledge limitations (restricted to in vitro regression datasets, lack of categorical outputs, no in vivo extrapolation) and outline future directions, such as QIVIVE extension, categorical data handling, and integration with broader drug discovery. This reflective treatment increases the transparency and usefulness of the work for the community.

**Weaknesses:**

1. **Unclear benefit of joint modeling**: The claimed advantage of jointly modeling PK and DTI is not substantiated. The paper does not include a comparison against task-specific baselines (a PK-only model and a DTI-only model trained with the same dataset). Without such ablations, it is unclear whether joint training provides real benefits beyond what could be achieved with two separate models, especially since the encoders for SMILES and proteins could also be trained independently (or replaced with strong pretrained models). This weakens the central claim of addressing "data overlap sparsity."
2. **Limited baseline coverage**: The evaluation only compares against cGAN and Syngand, which are not state-of-the-art. For a fairer assessment, the method should be benchmarked against more recent and task-specific baselines. For example, GraphormerDTI for drug–target interaction, and PEMAL for pharmacokinetic properties. Without these, the empirical superiority of xImagand-DKI remains unconvincing.
3. **Potentially misleading "generative" framing**: While the model is diffusion-based, the evaluation tasks focus on predicting PK and DTI properties. From the reader’s perspective, this looks closer to conditional property prediction than generative modeling in the usual drug discovery sense (e.g., novel molecule generation). The title and framing may therefore overstate the novelty of the contribution.
4. **Ambiguity in "domain knowledge infusion"**: The paper emphasizes "domain knowledge infusion," but what is actually infused are learned embeddings from data sources. While useful, these are standard feature engineering or multi-view representations, not incorporation of human expert knowledge or mechanistic rules. A more accurate description might be "multi-modal feature infusion" rather than domain knowledge, to avoid overstating the conceptual contribution.

**Questions:**

1. **Clarification on Figure 4**: The purpose of Figure 4 is not entirely clear. For the off-diagonal subplots, what exactly do the cross-property comparisons represent? In addition, some panels appear dominated by real data (orange) while others by synthetic data (blue). Could the authors clarify what these differences signify and how to interpret them?
2. **Interpretation of Table 5 results**: The results in Table 5 seem relatively weak for practical use. For instance, Pearson Correlation Coefficients below 0.5 are generally considered unreliable for downstream applications. Could the authors provide further explanation or justification for why these results should still be regarded as meaningful?
3. **Handling of noisy PK/DTI data** Both PK and DTI datasets are known to be noisy due to heterogeneous experimental conditions and measurement variability. How does the proposed method account for such uncertainty? Is there any mechanism (e.g., uncertainty modeling, robustness analysis, or data quality filtering) incorporated into the framework?

---

### Official Review · Reviewer_4YFr · 2025-10-20

**Soundness:** 3
**Presentation:** 3
**Contribution:** 2
**Rating:** 2
**Confidence:** 5

**Summary:**

This paper proposes xImagand-DKI, a diffusion-based framework that generates synthetic pharmacokinetic (PK) and drug-target interaction (DTI) data conditioned on molecular (SMILES) and protein representations to address data overlap sparsity in drug discovery. The model integrates domain knowledge by infusing molecular fingerprints and Gene Ontology-based protein embeddings, enhancing biological and chemical representation learning. Experimental results show that xImagand-DKI produces synthetic data that closely match real PK/DTI distributions and improves downstream prediction tasks compared to baselines such as cGAN and Syngand. The authors conclude that this approach can substantially reduce experimental costs and accelerate pre-clinical drug discovery.

**Strengths:**

- This paper addresses an important research problem by aiming to reduce the costs of in vitro screening and enable polypharmacy and drug combination studies, demonstrating translational potential for real-world pharmaceutical applications.

**Weaknesses:**

- While the paper emphasizes data overlap sparsity in PK and DTI datasets, it does not sufficiently explain why this challenge is fundamentally difficult or how it differs from general missing-data or multimodal fusion problems.

- Although the paper claims translational potential, it lacks detailed analysis on how the proposed synthetic data would concretely improve downstream drug discovery pipelines or clinical decision-making.

- The motivation is not clearly articulated. Although the paper identifies data overlap sparsity between PK and DTI datasets as a key issue, it does not clearly explain why this sparsity critically hinders drug discovery or why synthetic data generation is a preferable solution over existing approaches such as data imputation or multimodal fusion. Moreover, it remains unclear how the proposed framework directly alleviates this problem.

- The technical contribution is relatively incremental. The proposed xImagand-DKI primarily integrates pre-trained models (ProtBERT, ChemBERTa) and established embedding techniques (PO2Vec, FPFormer) within a standard conditional diffusion framework. The work lacks substantial algorithmic innovation or theoretical justification for this integration. Moreover, the key components including diffusion-based generation, Gene Ontology for protein representation, and SMILES/fingerprint-based molecular representation are all existing ideas, and this work mainly presents a straightforward combination of them.

- This paper lacks ablation on fusion strategy or conditioning mechanism. The authors didn't conduct a detailed exploration of how fusion choices (e.g., concatenation, attention-based fusion) or conditional embeddings affect performance, leaving unclear which component drives the observed improvements.

- The experiments only compare against older baselines (Syngand, cGAN) and do not include recent state-of-the-art diffusion or multimodal generative models for drug discovery, limiting the credibility of performance claims.

- The results are reported as mean values without standard deviations, confidence intervals, or statistical tests, making it difficult to assess robustness.

- Experiments rely solely on TDCommons and BindingDB datasets. The model’s generalizability to other datasets or unseen molecular/protein domains remains untested.

- There is no real-world validation or expert evaluation. The study does not include validation of the generated PK/DTI data by domain experts or experimental comparison with laboratory measurements, which weakens claims about practical utility.

- The evaluation focuses on distribution similarity, not utility. While the synthetic data statistically resemble real data, there is no demonstration that they improve performance on realistic downstream tasks such as drug repurposing, lead optimization, or molecular docking.

- There is no visualization or qualitative analysis of biological plausibility. The paper lacks visualization or case studies demonstrating that generated PK or DTI properties align with known biochemical trends or mechanistic insights.

- Code is not provided for reproducibility purposes.

**Questions:**

- Could the authors elaborate on why limited overlap between PK and DTI datasets poses a unique challenge beyond standard missing data or multimodal alignment problems? How does this sparsity concretely hinder downstream drug discovery tasks?

- Why was a diffusion-based approach selected over other generative frameworks such as VAEs, energy-based models, or graph-based imputers (e.g., graph representation for molecules)? Are there particular characteristics of PK/DTI data that make diffusion models especially well-suited to this problem?

- Given that diffusion-based molecular generation, multimodal pre-training, and ontology-based embeddings are well-studied, what is the specific novel aspect of xImagand-DKI beyond integrating these components? Is there any theoretical or architectural insight that explains why combining these modalities improves generation performance?

- The experiments mainly focus on statistical similarity (Hellinger distance, correlations). Could the authors provide evaluations showing that synthetic data improves real-world downstream tasks, such as drug repurposing or PK/DTI prediction?

---

### Official Review · Reviewer_nab7 · 2025-10-31

**Soundness:** 1
**Presentation:** 1
**Contribution:** 1
**Rating:** 0
**Confidence:** 3

**Summary:**

This work proposes a diffusion model for imputing missing pharmacokinetic and drug-target interaction labels.
The proposed model is shown to compare favourably against a conditional GAN and another diffusion model.

**Strengths:**

- The task of data-imputation in the realm of pharmacokinetics and drug-target interactions appears like it could be relevant.

**Weaknesses:**

- There are no error bars or statistical tests even though results are often very close.
   Furthermore, the table uses bold typesetting for "best" results very selectively.
   This is especially striking given the claim of _significant_ improvements on line 443.
 - The explanations are mostly unclear.
   I do not understand why there are two subfigures in Figure&nbsp;2 or what the difference is supposed to be.
   It is unclear how the different reachability modes of PO2Vec are used.
   It is not specified how the different embeddings are combined.
   E.g. on line&nbsp;233 mentions "integration", without further details.
 - It is unclear how competitive the baseline performance is,
   given that there are no references to established literature.
   Furthermore, the appendix suggests the baselines are (zero-shot?) linear probing,
   which is seldomly a reasonable baseline.
 - The experiments do not sufficiently show that the generated data is meaningful.
   Label distributions can be matched by sampling from the empirical distribution,
   but this does not imply that the generated labels match to the compounds at hand.
   It would be more useful to show that the model is able to generate correct labels
   for compounds with labels that have not been included in the training data.
 - The proposed model seems to be a combination of relatively standard ML components.
   In that sense, there is little novelty from a ML perspective,
   making it not well suited for the audience that ICLR aims to target.

### Minor Issues
 - DDPM citation incorrectly uses first name.
 - Diffusion hyperparameters are given before section&nbsp;3.1 introduces diffusion models.
 - Could it be that there is something off with the Diffusion formulas?

**Questions:**

I am afraid that the paper would require more changes than can be expected in a rebuttal.
However, I would need answers to the following questions:
 1. What is the state-of-the-art on prediction of PK (ADMET) labels?
 2. What is the state-of-the-art on prediction of DTI labels?
 3. Is there any evidence for significance of results (accounting for randomness)?
 4. How are the features combined?
 5. How do you verify that generated labels fit the molecules?
 6. Where lies the ML novelty of the proposed approach?

---

### Meta-Review · Area_Chair_dndF · 2026-01-05

**Summary:**

The paper introduces xImagand-DKI, a conditional diffusion-based framework designed to address "data overlap sparsity" in drug discovery—a scenario where pharmacokinetic (PK) and drug-target interaction (DTI) datasets share few common molecules. The authors propose generating synthetic labels for these missing entries by conditioning a diffusion model on molecular SMILES and protein sequences. To enhance performance, they utilize "Domain Knowledge Infusion" (DKI) by incorporating Gene Ontology (GO) embeddings and molecular fingerprints. While the goal of reducing experimental costs through synthetic data is well-motivated, the reviewers found the technical execution and empirical validation insufficient for acceptance.

**Reviewer Concerns:**

The reviewers' primary concerns center on a significant lack of technical novelty and empirical rigor. They characterized the model as an incremental combination of existing pre-trained architectures—such as ProtBERT and ChemBERTa—within a standard diffusion framework, without providing new theoretical insights or unique architectural innovations. Methodologically, the paper was criticized for failing to include error bars, standard deviations, or statistical significance tests, which is particularly problematic given that the reported improvements are minimal and some metrics, such as $R^2$, are even negative. Furthermore, the evaluation was deemed insufficient because it relies on outdated baselines and focuses on statistical distribution matching rather than demonstrating practical utility in downstream drug discovery tasks. These issues are compounded by a lack of clarity in the fusion strategy and a failure to provide code, which severely limits the work's reproducibility and biological credibility.

**Reviewer Scores:**

The authors remained inactive during the rebuttal and discussion period. I reckon the reviewers won't have any motivation to change the score.

---

### Decision · Program_Chairs · 2026-01-26

Reject